Genome-wide analysis of lectin receptor-like kinases family from potato (Solanum tuberosum L.)

Zhang Weina 1
Chen Zhongjian 2
Kang Yichen 1
Fan Yanling 1
Liu Yuhui 3
Yang Xinyu 1
Shi Mingfu 1
Yao Kai 1
Qin Shuhao 1 qinsh@gsau.edu.cn
1 College of Horticulture, Gansu Agricultural University , Lanzhou , China
2 Agro-biological Gene Research Center, Guangdong Academy of Agricultural Sciences , Guangzhou , China
3 Gansu Key Laboratory of Crop Improvement and Germplasm Enhancement, Gansu Agricultural University , Lanzhou , China
Kalendar Ruslan
Electronic publication date: 2020 Jun 10
Publication date: 2020
Volume: 8
Electronic Location ID: e9310
Received 2019 Oct 7; Accepted 2020 May 17
Copyright: © 2020 Zhang et al.
Copyright year: 2020
Copyright holder: Zhang et al.
License: This is an open access article distributed under the terms of the Creative Commons Attribution License, which permits unrestricted use, distribution, reproduction and adaptation in any medium and for any purpose provided that it is properly attributed. For attribution, the original author(s), title, publication source (PeerJ) and either DOI or URL of the article must be cited.
License URL: https://creativecommons.org/licenses/by/4.0/

Keywords: Potato, Lectin receptor-like kinase (LecRLKs), Gene duplication, Expressional pattern, RNA-seq-based-transcriptomics, Fungal disease resistance

Funding: Gansu Agricultural University GAU-XKJS-2018-225 National Natural Science Foundation of China 31260311 Natural Science Foundation of Gansu Province 1606RJZA034 China Postdoctoral Science Foundation 2012M512042 and 2014T70942 Ministry of Agriculture CARS-09-P14 This research was supported by Discipline construction fund project of Gansu Agricultural University (GAU-XKJS-2018-225), the National Natural Science Foundation of China (31260311), the Natural Science Foundation of Gansu Province (1606RJZA034) the China Postdoctoral Science Foundation (2012M512042 and 2014T70942), and the Ministry of Agriculture (CARS-09-P14). The funders had no role in study design, data collection and analysis, decision to publish, or preparation of the manuscript.

==============================
Lectin receptor-like kinases (LecRLKs) are involved in responses to diverse environmental stresses and pathogenic microbes. A comprehensive acknowledgment of the family members in potato (Solanum tuberosum) genome is largely limited until now. In total, 113 potato LecRLKs (StLecRLKs) were first identified, including 85 G-type, 26 L-type and 2 C-type members. Based on phylogenetic analysis, StLecRLKs were sub-grouped into seven clades, including C-type, L-type, G-I, G-II, G-III G-IV and G-V. Chromosomal distribution and gene duplication analysis revealed the expansion of StLecRLKs occurred majorly through tandem duplication although the whole-genome duplication (WGD)/segmental duplication events were found. Cis-elements in the StLecRLKs promoter region responded mainly to signals of defense and stress, phytohormone, biotic or abiotic stress. Moreover, expressional investigations indicated that the family members of the clades L-type, G-I, G-IV and G-V were responsive to both bacterial and fungal infection. Based on qRT-PCR analysis, the expressions of PGSC0003DMP400055136 and PGSC0003DMP400067047 were strongly induced in all treatments by both Fusarium sulphureum (Fs) and Phytophthora infestans (Pi) inoculation. The present study provides valuable information for LecRLKs gene family in potato genome, and establishes a foundation for further research into the functional analysis.

Introduction

Unlike animals, plants lack the basic survival ability to escape from the danger of environmental fluctuations. However, receptor-like kinases (RLKs), as one of the largest receptors, allow the plants to communicate between cells and to interact with the environment (Li et al., 2018). RLKs also have an important role in response to biotic and abiotic stresses (Nazarian-Firouzabadi et al., 2019). Typical RLKs comprise an extracellular ligand binding domain in N terminus and an intercellular cytoplasmic kinase domain in C terminus, and the two regions are connected via a transmembrane domain (Vaid, Pandey & Tuteja, 2012). Based on sequence analysis of the variable extracellular domain and the kinase domain, a total of 747 RLKs have been categorized into 52 subfamilies (Dezhsetan, 2017). However, Lectin receptor-like kinase (LecRLK) family was first described early in 1996 (Herve et al., 1996). Previous study shows that there are a total of 75 and 173 LecRLKs in Arabidopsis and rice, respectively (Vaid, Pandey & Tuteja, 2012). Expression of the LecRLK proteins have been reported in Lombardy poplar (Nishiguchi et al., 2002), Oryza sativa (Chen et al., 2006), Nicotiana benthamiana (Kanzaki et al., 2008), Nicotiana tabacum (Sasabe et al., 2007), Nicotiana attenuata (Gilardoni et al., 2011) Medicago truncatula (Navarrogochicoa et al., 2003) and Pisum sativum (Joshi et al., 2010), although no LecRLK has been found in yeast and humans (Navarrogochicoa et al., 2003).

Depending on the variability of N terminal lectin domain, LecRLKs has been classified into three types: G-, L- and C-type (Shiu & Bleecker, 2001). G-type LecRLKs possess α-d-mannose specific plant lectins. Many G-type LecRLKs also contain a cysteine-rich epidermal growth factor (EGF) domain and/or a plasminogen/apple/nematode (PAN) domain. However, EGF and PAN domains are only found in the G-type LecRLKs. The L-type LecRLKs contain a characteristic legume-lectin domain, which are similar with soluble lectin proteins found in leguminous plants (Herve et al., 1999). The C-type LecRLKs are thought to be homologs of calcium-dependent lectin domain. Previous study has identified 32 G-type and 42 L-type LecRLKs in Arabidopsis and 100 G-type and 72 L-type members in rice. However, results showed that both Arabidopsis and rice have only one gene encoding C-type LecRLK protein (Cambi, Koopman & Figdor, 2005).

LecRLKs have been reported to play a diverse role in plant multiple development stages, including seed germination, senescence, responses to wounding and salinity (Vaid, Macovei & Tuteja, 2013; Vaid, Pandey & Srivastava, 2015). Moreover, LecRLKs have been believed to be involved in tolerance to biotic and abiotic stresses, especially in pathogen attack (Cheng et al., 2013; Singh & Zimmerli, 2013; Liu et al., 2015; Wang et al., 2015), which is similar to the role of RLKs reported by Nazarian-Firouzabadi et al. (2019). Transcript-level analyses revealed variable gene expression responses to diverse abiotic stresses, such as salt, drought, wounding and extreme temperatures (Vaid, Pandey & Tuteja, 2012; Vaid, Macovei & Tuteja, 2013). The inhibition of LecRK-b2 can result in salt- and osmotic stress-insensitive root elongation (Deng et al., 2009). The overexpression of Arabidopsis thaliana L-type lectin-like protein kinase 1 (AtLPK1) can enhance seed germination and cotyledon greening under high salinity conditions (Huang et al., 2013). The overexpression of Glycine sojaLec RLK (GsSRK) and Pisum sativum LecRLK (PsLecRLK) can enhance salt tolerance (Vaid, Pandey & Srivastava, 2015; Sun et al., 2013). The LecRLKs are also considered to be potential plant immune receptors, which play fundamental role in microbe- and plant-derived molecular patterns associated with pathogen defense as the key pattern-recognition receptors (Singh & Zimmerli, 2013; Wang et al., 2015; Wu & Zhou, 2013; Huang et al., 2014). Several LecRLKs played role in conferring resistance to rice against fungal pathogen (Chen et al., 2006) and bacterial pathogen (Singh & Zimmerli, 2013).

Potato (Solanum tuberosum L.) is considered as one of the valuable food security crops in the word, with a global production of 375 million tons in 2018 (National Potato Council, 2018). The cultivated potato ranks fourth food crops planted and consumed worldwide after rice, wheat and maize (FAOSTAT, 2011). China, India, and Russia are principal potato producers (National Potato Council, 2018). However, because of the spread of pathogenic microorganisms, crop capacity and quality of potatoes produced by these countries are far behind those in developed countries. Date showed that the total annual potato loss due to bacteria/fungi and viruses worldwide is 14% and 7%, respectively (Oerke, 2006). In Gansu province of China, late blight and dry rot, caused by Phytophthora infestans (Pi) and Fusarium sulphureum (Fs), respectively, result in huge losses in yield (He et al., 2004; Hui et al., 2010).

With the development of high-throughput sequencing technology, the transcriptome of potato and some other plants have been published, which have laid a foundation for the identification of large gene families. Besides Arabidopsis and rice, LecRLKs gene have been identified in different plants including Populus, soybean, shrub, moss and Eucalyptus (Yang et al., 2016). However, limited information is available about LecRLKs in potato. In the current study, we firstly conducted a comprehensive bioinformatics analysis of the LecRLKs in potato genome. The evolutional characteristics were investigated through phylogenetic relationship, chromosomal distribution and gene duplication events. Moreover, based on the published expression data and quantitative real-time PCR (qRT-PCR), the expressional patterns of the family members were analyzed during potato against to biotic stresses, which provided a novel insight for the functional analysis of the LecRLK gene family in potato.

Methods

Annotation of LecRLKs

The amino acid and nucleotide sequence data of Potato (Solanum tuberosum) v3.4, Arabidopsis thaliana Araport11, Rice (Oryza Sativa Japonica Group) IRGSP-1.0, Apple (Malus domestica) GDDH13 v1.1, and Tomato (Solanum lycopersicum) SL2.50 were downloaded from the Potato Genome Sequencing Consortium (PGSC) Public Data (http://solanaceae.plantbiology.msu.edu/pgsc_download.shtml), The Arabidopsis Information Resource (TAIR) (https://www.arabidopsis.org), the Ensembl Genomes (http://ensemblgenomes.org/), the genome database for Rosaceae (GDR) (https://www.rosaceae.org/), and the Solanaceae Genomics Network (https://solgenomics.net/), respectively. Proteins contained kinase domain (PF00069) or Pkinase_Tyr (PF07714) in above five species were following the method described by Zhu et al. (2018). After that, sequences contained a N-terminal domain B_lectin (PF01453), Lectin_legB (PF00139) or Lectin_C (PF00059) were considered as LecRLKs. All of the candidate proteins were determined by using online software Pfam (http://pfam.xfam.org/) and SMART (http://smart.embl-heidelberg.de/) (E-value < 10−5) to ensure the presence of N-terminal lectin domain, a transmembrane region and C-terminal kinase domain in each putative member.

Multiple sequence alignment, phylogenetic analysis, physical and chemical properties and subcellular localization classification

The sequences of all LecRLKs from six species were aligned using ClustalX 2.0 with default parameters (Larkin et al., 2007) and a phylogenetic tree was constructed using the Neighbor-Joining (NJ) method in MEGA 7 (Kumar, Stecher & Tamura, 2016). The parameter partial deletion and P-distance Model were used with 1,000 bootstrap replicates. The protein theoretical molecular weight (MV) and theoretical isoelectric point (pI) were predicted using ProtParam (https://web.expasy.org/protparam/) (Gasteiger et al., 2005). The subcellular localization of proteins was predicted using the WOLF PSORT (https://www.genscript.com/wolf-psort.html) and CELLO v2.5: subcellular Localization predictor (http://cello.life.nctu.edu.tw/).

Analysis of chromosomal distribution, gene duplication and Gene Ontology overview

The chromosomal positions of potato LecRLKs (StLecRLKs) were obtained from PGSC (http://solanaceae.plantbiology.msu.edu/pgsc_download.shtml) and the chromosomal distribution were visualized by using software MapDraw 2.1 (Liu & Meng, 2003). Tandem duplicated genes are defined as those genes that (1) are ≤l0 genes apart, (2) belong to the same phylogenetic group, and are (3) within 100 kb (Hofberger et al., 2015). The Whole-Genome Duplication (WGD)/segmental duplication was calculated by MicroSyn software (Cai et al., 2011). The projected GO annotation for the candidate StLecRLK genes was then analyzed for enrichment of GO terms using GOEast with default settings (Zheng & Wang, 2008). For cis-acting element analysis, genomic DNA sequences in the promoter region (−1,500 bp) were downloaded from PGSC and then scanned for in the PlantCARE database (http://bioinformatics.psb.ugent.be/webtools/plantcare/html/) (Lescot et al., 2002).

Expression patterns of StLecRLKs in response to biotic infection

To investigate the responses of StLecRLKs to biotic stress, expression data were obtained from the published studies, including potato infected with pathogenic bacteria Candidatus Liberibacter solanacearum (Cls) (Levy et al., 2017), Pectobacterium carotovorum subsp. brasiliense (Pcb) (Kwenda et al., 2016) and Ralstonia solanacearum (Rs) (Zuluaga et al., 2015), fungi Phytophthora infestans (Pi) (Yogendra & Kushalappa, 2016), and Fusarium sulphureum (Fs) (He et al., 2018). The expression pattern was extracted by the accession number of StLecRLKs. False Discovery Rate (FDR) < 0.05 and |log2 (fold change)| ≥ 1 were used as a threshold to judge the significance of the difference in gene expression. The heatmap was visualized by using software Multi experiment viewer software (MeV 4.9.0) (http://mev.tm4.org/).

Plant materials, treatment and gene expressional assay

Potato cultivar Helan 15 (Favorita, Qingshenshu 2007001) were cultured in the greenhouse with 24 °C. After 5 weeks cultivation, leaf tissues were collected, sterilized with 75% ethanol and transferred into a petri dish with a diameter of 9 cm. To maintain humidity, a wet filter paper was placed in each petri dish. Pi strain “HB09-16-2 (race 1.2.3.4.5.6.7.9.10.11, collected from Hubei Province, China)” were friendly supplied by Prof. Tian Zhendong, Huazhong Agricultural University. The pathogen was cultured at 19 °C for 2 weeks. The conidia were collected and adjusted to a concentration as 7 × 104 sporangia per milliliter. Inoculation was performed following the method described by He et al. (2018). For the control, the conidia suspension was replaced with the same volume of sterilized water. Inoculated and control tissues were collected in triplicate at 1 d, 4 d, 6 d, and 8 d after inoculation and quick-frozen in liquid nitrogen and stored at −80 °C.

The method of inoculation of Fs was performed as described by Li et al. (2009). Longshu No. 7 (Guoshenshu 2009006), without physical injuries or visible infection, was used as the material for the inoculation for 1 d, 2 d, 3 d and 4 d. Before treatment, tubers were surfaced-disinfected with 2% sodium hypochlorite for 3 min, and then rinsed with tap water and air-dried.

RNA isolation, reverse-transcription and qRT-PCR were performed following the method described by Zuo et al. (2018). Sequences of primers and actin genes could be found out in File S1. The relative expression levels of each gene were calculated based on the 2−ΔΔCT method (Livak & Schmittgen, 2001).

The above samples at different treatment time were collected from three independent replicates. Each experimental unit consisted of three plants or tubers, three leaves per plant and each leaf or tuber with one inoculation site. The data were statistically analyzed using software Origin 8.0 and the significant differences were analyzed using the t-test (*P < 0.05; **P < 0.01).

Results

LecRLKs annotation and classification

Based on domain screening, a total of 113 transcripts encoding LecRLKs were identified in potato, including 85 G-type, 26 L-type and 2 C-type members (File S2). For some transcripts, two or more transcripts had the same genome loci and corresponded with the same gene accession, but most of these showed the distinct sequences. Therefore, we determined the number of LecRLKs as the transcript’s numbers. With the same criteria, 80, 76, 139 and 167 LecRLKs were annotated in Tomato (Solanum lycopersicum), Arabidopsis, Rice (Oryza sativa) and Apple (Malus domestica), respectively. From the published study, we also re-identified the LecRLKs in different plant species, including Populus, Eucalyptus, Shrub, Soybean, Corn and Moss (Table 1; Yang et al., 2016). To ensure the uniformity of the obtained data, domains of all sequences were determined based on the same criterion as above. We excluded several proteins which did not contain typical domains of LecRLKs (File S2). Compared to Moss, the member of G-type and L-type LecRLKs was much larger in other plants, and the number of G-type LecRLKs was exceed that of L-type LecRLKs except for Shrub and Arabidopsis. There were only one or two C-type LecRLKs in all the detected plants. Over half of LecRLKs were classified as G-type except in Shrub, Arabidopsis, Corn and Moss. In potato, tomato, apple and Populus, G-type LecRLKs accounted for more than 70% of the LecRLKs, which suggested that the G-type LecRLKs were rapidly expanded in potato, tomato and Populus than in the other detected plants.

Table 1 The number of G-type, L-type and C-type LecRLKs in 11 plant species.

Plant species	Sub-group	Total	
G-type	L-type	C-type	
Solanum tuberosum	85 (75.22%)	26 (23.01%)	2 (1.77%)	113	
Solanum lycopersicon	60 (75%)	19 (23.75%)	1 (1.25%)	80	
Malus domestica	118 (71.08%)	46 (27.71%)	2 (1.2%)	166	
Populus	180 (77.92%)	50 (21.65%)	1 (0.43%)	231	
Eucalyptus	118 (59.6%)	79 (39.9%)	1 (0.51%)	198	
Shrub	25 (44.64%)	30 (53.57%)	1 (1.79%)	56	
Soybean	123 (65.08%)	64 (33.86%)	2 (1.06%)	189	
Arabidopsis thaliana	34 (44.74%)	41 (53.94%)	1 (1.32%)	76	
Oryza sativa	80 (57.55%)	57 (41.01%)	2 (1.44%)	139	
Corn	46 (48.42)	48 (50.53%)	1 (1.05%)	95	
Moss	2 (40%)	1 (20%)	2 (40%)	5	

To further investigate the classification and evolution relationships of LecRLKs, phylogenetic tree of three independent types (G-type, L-type and C-type) were constructed separately based on the multiple sequence alignment of the full-length amino acid sequences from five plant species, including potato, tomato, apple, Arabidopsis and rice. For G-type LecRLKs, the members could be subdivided into five clades (G-I to G-V) (Fig. 1). Clades G-I, G-II, G-IV and G-V contained LecRLKs family members from all the tested plants, while G-III only contained members from rice. Potato G-type LecRLKs were discovered from four clades except G-III. There were 28, 3, 25 and 29 members in G-I, G-II, G-IV and G-V clade which accounted for 25.93%, 7.14%, 25% and 26.61% in each subgroup respectively.

Figure 1 Phylogenetic analysis of G-type LecRLKs from five different plants.

The phylogenetic tree was generated from the alignment result of the full-length amino acid sequences by the neighbor-joining (NJ) method. All StLecRLKs members, together with homologs of Arabidopsis thaliana, apple (Malus domestica), rice (Oryza sativa) and tomato (Solanum lycopersicum), were classified into five distinct clades. LecRLK subgroups were shown in different colors.

L-type LecRLKs was subdivided into eight clades (Fig. 2). L-V had two Arabidopsis thaliana LecRLKs. L-VI contained members from four investigated plant species except rice. Other subgroups contained members from all five species. Potato LecRLKs located in six clades except L-I and L-V. C-type LecRLKs could be divided into four subgroups (Fig. S1), in which subgroup IV contained members from potato and tomato. For the other subgroups, each of these only contained members from one single plant species. Taken together, these results indicated the distinctly evolutional process of LecRLKs in plants. The division of each clade was supported by high bootstrap values.

Figure 2 Phylogenetic analysis of L-type LecRLKs from five different plants.

The phylogenetic tree was generated from the alignment result of the full-length amino acid sequences by the neighbor-joining (NJ) method. All StLecRLKs members, together with homologs of Arabidopsis thaliana, apple (Malus domestica), rice (Oryza sativa) and tomato (Solanum lycopersicum), were classified into eight distinct clades. LecRLK subgroups were shown in different colors.

StLecRLKs classification, domain organization, physicochemical properties and gene ontology analysis

To further confirm the classification and evolutionary relationship of StLecRLKs, phylogenetic tree generated from the alignment result of the full-length amino acid sequences (Fig. 3; File S3). StLecRLKs could be divided into seven clades, including C-type, L-type, G-I, G-II, G-III, G-IV and G-V, which contained 2, 26, 28, 3, 8, 15 and 31 members, respectively.

Figure 3 Phylogenetic analysis of LecRLKs in potato (Solanum tuberosum).

The phylogenetic trees were constructed using MEGA 7 software. The numbers were bootstrap values based on 1,000 iterations. Note that three different types of StLecRLKs were categorized clearly in seven clades. Different color was used to distinguish different clades.

Domain organization was also investigated for each subgroup of StLecRLKs (File S2). Unique N-terminal domain was discovered from both L-type (Lectin_legB) and C-type members (Lectin_C). However, complex domain composition was found from G-type LecRLKs. Besides B_lectin domain, multiple additional N-terminal domains were detected, such as S_locus_glycop (PF00954), PAN_2 (Pfam accession number was not detected) and EGF (SM000181). Additionally, two intracellular domains, DUF3660 (PF12398) and DUF3403 (PF11883) were observed from some members. Domain DUF3403 was in C-terminal of specific protein, but its function is still unknow. Above all, G-type StLecRLKs, being the largest group, showed a more complex evolutionary relationship and domain organization than the other types.

After determining the conserved domains of StLecRLKs, the physicochemical properties and subcellular localization of these proteins were also analyzed (File S4). The results showed that 113 StLecRLKs had an amino acid size ranging from 419 to 1,504 aa, a molecular weight ranging from 47.07 to 169.51 kDa, and pI value ranging from 5.02 to 9.26. The subcellular localization of 113 StLecRLKs were predicted by the online software CELLO and PSORT. 87 StLecRLKs were located in the plasma membrane, two in the cytoplasm, one was in the extracytoplasmic surface and three in both plasma membrane and cytoplasm. Moreover, there were 20 StLecRLKs with inconsistent prediction results from the two software. Further, 15 and 20 GO terms were distributed into molecular function and biological process (File S5). Within biological process, a larger number of family members involved in “metabolism process”, “cell recognition”, and “protein modification process”. For the molecular function categories, genes were mainly associated with “kinase activity”, “catalytic activity”, “transferase activity” and “binding”.

Chromosomal distribution, gene duplication

Tandem duplication mechanism represents a common role in the expansion of RLK family. To identify the gene clusters and to investigate such events, we scanned the gene ID and the corresponded protein ID of StLecRLKs based on PGSC database (File S2). For several accessions, unique gene accession was corresponded to two or more protein accessions. For instance, gene PGSC0003DMG400019237 encoded three proteins, including PGSC0003DMP400033418, PGSC0003DMP400033419 and PGSC0003DMP400033420. Finally, 88 of the 113 StLecRLKs was visualized on chromosome across 12 chromosomes (Fig. 4). There were 15, 10, 9, 20, 11 and 7 members scattered on Chr 02, 03, 04, 07, 09 and 10, respectively. Additionally, 15 tandem duplicated gene clusters (marked in red vertical lines) and five WGD/segmental duplicated gene pairs (marked in blue dotted lines) were found from StLecRLKs. The tandem duplicated gene clusters contained 47 StLecRLKsconsisting of 40 G-type (black) and 7 L-type (green). On the other hand, WGD/segmental duplicated gene pairs contained seven G-type and two L-type StLecRLKs. Above results indicated that tandem duplications played important roles in the amplification of the StLecRLK gene family.

Figure 4 Distribution of LecRLKs genes on potato (Solanum tuberosum) chromosomes.

Graphical representation of physical locations for each LecRLK on chromosomes. Tandem duplicated genes on the particular chromosome were depicted by red vertical lines. Chromosomal distances were given in Mb.

Cis-elements involved in StLecRLKs

In order to further explore the functions of the candidate StLecRLKs, a 1,500 bp promoter region was extracted and PlantCARE was used to analyze the cis-elements related to stress challenge. A great number of cis-elements responsive to defense and stress, phytohormines (abscisic acid (ABA), methyl jasmonate (MeJA) and salicylic acid (SA)), biotic stressors (elicitor) and abiotic stressors (drought, low temperature and wounding) were identified in the promoter region of StLecRLKs (Fig. 5; File S6). Of these, the number of genes which had the cis-element involved in ABA-, MeJA- and SA-responsiveness were 57, 57 and 34, respectively. Most of the StLecRLKs belonged to the phylogenetic group G-I, G-V and L-type. Additionally, some StLecRLKs had elements responsive to abiotic stress, and several members possessed elements responsible for elicitor. The above results indicated that StLecRLKs had a potential role in response to signals from phytohormones and multiple stresses (Fig. 5).

Figure 5 Cis-element detection in the promoter region of StLecRLK.

Signals of cis-element were responsive for stress challenge, including deference and stress, elicitor, low-temperature, drought, wounding and hormone (SA, ABA and MeJA). Different color was used to distinguish different clades.

Expression patterns of StLecRLKs in response to biotic stresses

Based on the published transcriptomic datasets, the expressional patterns of StLecRLKs were investigated during potato infected with phytopathogenic bacteria (Levy et al., 2017; Kwenda et al., 2016; Zuluaga et al., 2015) and fungi (Yogendra & Kushalappa, 2016; He et al., 2018). For the bacteria, expression changes of StLecRLKs were extracted after potato inoculated with Cls, Pcb and Rs (Fig. 6). A total of 55 StLecRLKs were differently expressed, including 15 members in G-I, 2 in G-III, 10 in G-IV, 17 in G-V and 11 in L-type, respectively. Furthermore, 28 StLecRLKs were detected to be up-regulated under at least one bacterial infection.

Figure 6 Expression patterns of StLecRLKs in response to phytopathogenic bacteria.

The heatmap showed the differentially expressed genes of Waneta potato infected with Candidatus Liberibacter solanacearum (Cls), Solanum tuberosum cv. Valor compared with S. tuberosum cv. BP1 at each time-point inoculated with Pectobacterium carotovorum ssp. brasiliense (Pcb), and differentially-expressed genes in inoculated potato roots with Ralstonia solanacearum (Rs). The classification of genes in phylogenetic trees were listed next to the gene name. Yellow indicated genes that were up-regulated, blue indicated genes that were down-regulated, and white indicated genes that were not changed significantly.

Additionally, expression patterns of StLecRLKs were extracted from published transcriptomic data of potato tuber (Gao et al., 2013) and leaf (Yogendra & Kushalappa, 2016) inoculated with Pi (Fig. 7). A total of 26 members, including nine members in G-I (32.14%), two in G-III (25.0%), two in G-IV (13.33%), five in G-V (16.13%) and eight in L-type (30.77%) were differentially expressed (Fig. 7). PGSC0003DMP400030282 showed the same expressional pattern among resistant tuber and leaves. However, PGSC0003DMP400032146 and PGSC0003DMP400055136 in G-I and PGSC0003DMP400045382 in G-IV were detected to be down-regulated under bacteria attack while they were up-regulated under fungi infection. PGSC0003DMP400067047 in G-IV was down-regulated under both bacteria and fungi infection (Figs. 6 and 7). Combined with the above results, a great number of G-I, G-V and L-type StLecRLKs were differentially expressed after potato tissue inoculated with both phytopathogenic fungi and bacteria, which indicated the potential roles of these members in potato disease resistance.

Figure 7 Expression patterns of StLecRLKs in response to Phytophthora infestans (Pi).

The heatmap of differentially expressed genes between F06025 (derived from AWN86514-2 × N06993-13 cross) and F06037 (derived from frontier russet × N1503-19 cross) genotypes, RB (tubers of transgenic “Russet Burbank” line SP2211 potato) and WT (tubers of nontransformed “Russet Burbank” (WT) potato) inoculated with Pi. The classification of genes in phylogenetic trees were listed next to the gene name. Bins in yellow were significantly up-regulated; bins in blue were significantly down-regulated and bins in white did not change significantly.

Expression patterns of StLecRLKs during potato infected with Fs and Pi

To further validate the regulating roles of the candidate StLecRLKs to fungal disease resistance, the expression analysis of StLecRLKs during potato inoculated with Fs (Fig. 8) and Pi (Fig. 9) were detected. Two or three members in each phylogenetic group were selected, which were strongly affected under the biotic stress treatment (Figs. 6 and 7). Finally, 16 differently expressed members were selected to perform qRT-PCT analysis (File S7). During Fs infection, 10 of the 16 detected StLecRLKs were differentially expressed at least one time point (Fig. 8; File S7), and they were detected to be up-regulated after treatments. The expression of PGSC0003DMP400067047 was gradually upregulated during the experiment, and the expression level was 26.59 times that of the control group at 4 d treatment. When the leaves were inoculated with Pi, 12 representative genes were differentially expressed (Fig. 9; File S7). PGSC0003DMP400041968, PGSC0003DMP400049549 and PGSC0003DMP400055136 were significantly induced at the later stage (T4), and the expressional levels were 12.33, 430.80 and 34.08 times that of the control group. PGSC0003DMP400030282, PGSC0003DMP400001977 and PGSC0003DMP400045383 were significantly induced when the leaves were treated for 6 d (T3).

Figure 8 Expressions of 10 StLecRLKs during potato inoculated with Fusarium sulphureum (Fs).

(A–H) represent eight G-type LecRLK genes, in which (A–G) belonged to G-I clade and (H) was in G-III clade; (I–J) two L-type LecRLK genes. Samples at 1-, 2-, 3- and 4-days post Fs inoculation were collected. The qRT-PCR data presented here were from three independent biological replicates and error bars indicated the standard deviation (SD). The asterisks indicated a significant difference as compared with the group inoculated with sterilized water at each time point (*p < 0.05 and **p < 0.01).

Figure 9 Expressions of 12 StLecRLKs during potato inoculated with Phytophthora infestans (Pi).

(A–J) represent 10 G-type LecRLK genes, in which (A–F) belonged to G-I clade, (G) was in G-II clade and (H–J) were in G-III clade; (K–L) two L-type LecRLK genes. Samples at 1-, 2-, 3- and 4-days post Pi inoculation were collected. The qRT-PCR data presented here were from three independent biological replicates and error bars indicated the standard deviation (SD).The asterisks indicated a significant difference as compared with the group inoculated with sterilized water at each time point (*p < 0.05 and **p < 0.01).

Discussion

LecRLKs are considered as important regulators for external stimuli, such as environmental stresses and pathogen attack (Bellande et al., 2017). Previous literatures reported the identification, evolution and function of LecRLKs in several plant species (Vaid, Pandey & Tuteja, 2012; Yang et al., 2016; Zhao et al., 2016; Ma et al., 2018; Liu et al., 2018; Zhao et al., 2018). Currently, a comprehensive analysis of the LecRLK family in potato was carried out, and 113 LecRLK genes including 85 G-type, 26 L-type and 2 C-type members were identified and analyzed. Furthermore, investigations for classification, evolution, gene ontology, cis-elements and expression patterns revealed several important features of the family members.

Evolution and expansion of StLecRLKs

Besides the differences in genome size, expansion rate is one of the most important contributors to the copy numbers of plant LecRLKs (Liu et al., 2018). Compared to Moss, total numbers of LecRLKs were drastically expanded in higher plants (Table 1), which mainly resulted from the expansion of the G-type LecRLKs members, followed by L-type. In potato, tomato, apple and Populus, about three-quarter of LecRLKs were identified as G-type members, although L-type LecRLKs were more expanded than G-type members in several species such as Shrub, Arabidopsis and Corn (Table 1; Vaid, Pandey & Tuteja, 2012). The results showed that different plant species had different numbers of LecRLKs, which was possibly related with the different growth condition and growth behaviors, or maybe with the plant life cycles and the mode of reproduction. Yang et al. (2016) also speculated that the difference in the number of G-type and L-type LecRLKs between Arabidopsis and Populus may be related with the fact that Arabidopsis is a self-fertile plant whereas Populus is an obligate outcrossing plant. In our study, the number of LecRLK in Arabidopsis and rice is different from the previous researchers, which may be due to the perfection of genome sequence or the difference of identification standards (Table 1).

Phylogenetic analysis revealed the distinct evolutionary process in plant species (Figs. 1 and 2; File S1). The rice specific clade was found both in G-type (G-III) and L-type (L-I) LecRLKs. Therefore, we suggested the distinct expanded characteristics between G-type and L-type LecRLKs in plant species. Compare to L-type LecRLKs in potato, more G-type members were found, indicating the crucial roles of G-type LecRLKs in development and environmental response.

Gene duplications and functional analysis

Duplication can lead to functional divergence of genes, which is a universal phenomenon in plants (Liu et al., 2012). LecRLKs duplication were mainly resulted from tandem duplication and segmental duplication/WGDs (Yang et al., 2016; Hofberger et al., 2015; Shumayla Sharma et al., 2016). In potato, 40 G-type StLecRLKs were involved in tandem duplication but only seven members in WGD/segmental duplication (Fig. 4). Similarly, there were more tandem duplicated L-type StLecRLKs than that in segmental duplication/WGD (Fig. 4). Hence, the tandem duplication might be the main reason for expansion of both G-type and L-type StLecRLKs. Compared to RLKs involved in development, stress responsive RLKs showed a higher number of tandemly duplicated genes. Based on the above conclusion, we deduced that StLecRLKs also took important roles in potato responses to biotic and abiotic stress. Furtherly, we used GO method to analyze the molecular function of these genes. The results showed that the enriched GO terms were kinase activity, catalytic activity, transferase activity and binding (phosphate binding and carbohydrate binding) (Fig. 5). Since chitin and β-1,3 glucans are components of the cell walls of some higher fungi, some plant lectins sharing chitin-binding domains can recognize the signal then naturally these substrates will be hydrolyzed and fungal growth will be disrupted (Sahai & Manocha, 1993). This type of lectin could inhibit the growth of several phytopathogenic and saprophytic chitin containing fungi in vitro (Broekaert et al., 1989). Moreover, the cis-element prediction furtherly demonstrated that StLecRLKs have the potential role against multiple stresses (Fig. 5; File S6). Previous study confirmed LecRLKs participated in the regulation of abiotic stress (Vaid, Pandey & Srivastava, 2015; Singh & Zimmerli, 2013; Xin et al., 2009) and biotic stress (Chen et al., 2006; Wang et al., 2015).

Complex domain architecture and organization of G-type StLecRLKs

Protein domains are some complex regions of a protein’s structure, which often perform some specific function (Buljan & Bateman, 2009). Architecture and composition of domain of G-type StLecRLKs were observed (File S2). In addition to B_lectin domain, most G-type members contained an S_locus_glycoprotein, EGF and PAN_2 or PAN-AP domain, in which PAN domains are involved in interactions of protein–protein and protein–carbohydrate (Naithani et al., 2007). Furthermore, some G-type StLecRLKs also contained intracellular domain DUF3660 or DUF3403. Our result was consistent with the findings in Populus although the function of the two domains was largely limited until now (Yang et al., 2016). Nevertheless, the complex domain architecture suggested the diverse functions of G-type StLecRLKs.

StLecRLKs in response to biotic stresses

Increased investigations reported the crucial role of both G-type and L-type LecRLKs in plant immunity (Zhao et al., 2018; Wang et al., 2015, 2014). Over-expression of LecRK-V (L-type) significantly enhanced the resistance of Yangmai158 to powdery mildew (Wang et al., 2018). Similar studies have been done in rice and tomatoes. Pid2 (G-type LecRLK in rice) and NbLecRK (L-type LecRLK in tomato) were shown to increase the resistance to fungal pathogen and Phytophthora, respectively (Chen et al., 2006; Wang et al., 2015). In this study, we found a large number of StLecRLKs were differentially expressed during potato infected both bacterial and fungal pathogens (Figs. 6 and 7). Most of these were distributed in clades G-I, G-V and L-type. Genes which contributed to plant resistance to a wide of pathogens could be classified as broad-spectrum resistant genes (Dangl & Jones, 2011; Xiao et al., 2004). In Populus, the root-expressed PtLecRLKs maybe have the potential function for perceiving signals from soil microbes (Yang et al., 2016). Intriguingly, the expression of PGSC0003DMP400030282 was induced under signals from both phytopathogenic bacteria and fungi, which suggested that the crucial roles in broad-spectrum disease resistance of potato, while PGSC0003DMP400055136 and PGSC0003DMP400067047 showed distinct expression, indicating a different role in response to different pathogen signal (Figs 7–9).

Conclusions

In conclusion, a systematic study was carried out to identify and characterize the LecRLK family genes in potato. The phylogenetic analysis showed the distinctly evolutionary processes of the family members in different plants. G-type StLecRLKs exhibited a rapid enlargement of family members, followed by L-type members. The expansion of both G-type and L-type StLecRLKs was mainly resulted from the tandem duplication. In addition, a large number StLecRLKs were differentially expressed in response to bacteria and fungi. Among these, PGSC0003DMP400055136 and PGSC0003DMP400067047 were strongly induced by both Fs and Pi, which could be used as candidate genes for functional analysis of resistance. Our study provides insight into further research on the functions and mechanisms of this important RLK subfamily.

Supplemental Information

Supplemental Information 1 Phylogenetic analysis of C-type LecRLKs from Arabidopsis thaliana, apple (Malus domestica), potato (Solanum tuberosum), rice (Oryza sativa) and tomato (Solanum lycopersicum).

Different color was used to distinguish different subgroups. The neighbor-joining (NJ) method was used to analyze the evolutionary trees.

Click here for additional data file.

Supplemental Information 2 qRT-PCR primer sequences for actin and the selected potato genes.

Click here for additional data file.

Supplemental Information 3 List of LecRLKs in different plants.

Click here for additional data file.

Supplemental Information 4 The full alignment of 113 StLecRLK proteins.

Click here for additional data file.

Supplemental Information 5 The physical and chemical properties and prediction of subcellular localization of StLecRLKs.

Click here for additional data file.

Supplemental Information 6 Gene ontology enrichment analysis of StLecRLKs.

Click here for additional data file.

Supplemental Information 7 Cis-elements in the promoter region of StLecRLKs.

Click here for additional data file.

Supplemental Information 8 The raw data of the qRT-PCR.

Click here for additional data file.

We would like to thank Prof. Xiong Xingyao (The Institute of Vegetables and Flowers Chinese Academy of Agricultural Sciences, Beijing, China) for providing Helan 15 potatoes. We also would like to thank Dr. Bai Lijun and Xu Tong (Chengdu Life Baseline Technology Co., Ltd., Chengdu, China) for technical assistance with RNA sequencing and bioinformatic analysis.

Additional Information and Declarations

Competing Interests

Author Contributions

Data Availability

The authors declare that they have no competing interests.

Weina Zhang conceived and designed the experiments, performed the experiments, analyzed the data, prepared figures and/or tables, authored or reviewed drafts of the paper, and approved the final draft.

Zhongjian Chen conceived and designed the experiments, prepared figures and/or tables, and approved the final draft.

Yichen Kang performed the experiments, prepared figures and/or tables, and approved the final draft.

Yanling Fan performed the experiments, prepared figures and/or tables, and approved the final draft.

Yuhui Liu performed the experiments, prepared figures and/or tables, and approved the final draft.

Xinyu Yang analyzed the data, authored or reviewed drafts of the paper, and approved the final draft.

Mingfu Shi analyzed the data, authored or reviewed drafts of the paper, and approved the final draft.

Kai Yao analyzed the data, authored or reviewed drafts of the paper, and approved the final draft.

Shuhao Qin conceived and designed the experiments, analyzed the data, authored or reviewed drafts of the paper, and approved the final draft.

The following information was supplied regarding data availability:

Data is available at the Sequence Read Archive: SRS4823473.

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
