# Peer review of "Genome-wide analysis of lectin receptor-like kinases family from potato (Solanum tuberosum L.)"

_PeerJ, doi:10.7717/peerj.9310_

## Round 0.1 · original submission · Major Revisions

Authors need to prepare a new improved version of the article taking into account the recommendations of reviewers.

The language has to be checked throughout the manuscript by a native speaker.

Reviewer 1 ·

Basic reporting

Manuscript should be modified in some parts and English improved, fgure are not clear and not well labeled

Experimental design

The work can give a good contrbiture to the research area. However statements made should be carefully revised. It is important descrive how was made data collection to understand if genomes have the same coverage, similar annotation etc

Validity of the findings

Comparative Analysis should not be performed with data realesed from other authors. The analysis should be performed with omegeneus datasets

Additional comments

Zhang and Quin have submitted a manuscript entitled ‘Genome-wide analysis of lectin receptor-like kinases family from potato (Solanum tuberosum L.)’ to be considered for publication in J Peer. Authors annotated LecRLKs members in potato investigating their diversification. The expansion of family, occurred mainly by tandem duplication, and chromosome regions containing LecRLK members were highlighted. Transcriptomic investigations of potato infected with bacteria and fungi revealed that LecRLKs belonging to L-type, G-I, G-IV and G-V are high responsive. In particular qRT-PCR profiles of potato infected with Fusarium sulphureum (Fs) or Phytophthora infestans (Pi), indicated the activation of two common members. The effort made to analyze this gene family was useful. However, some important issues should be resolved before paper can be considered for publication. I suggest to revise carefully data obtained and to improve result presentation and discussion
Abstract
The sentences “Based on phylogenetic analysis of LecRLKs from potato, Arabidopsis thaliana, apple (Malus domestica), rice (Oryza sativa) and tomato (Solanum lycopersicon), distinct evolutional process of LecRLKs were found” should be deleted since data came from literature and they are original data produced in this paper .
Substitute 7 clusters with 7 clades
Based on qRT-PCR for potato incubated . It better rewording in Based on qRT-PCR for potato infected

Change: genes for functional analysis and further resistant breeding in genes for functional analysis and breeding.
Introduction
From line 24 to line 26. It is not clear why you highlight the function of C type if you not provide evidence in your work of their involvement in innate immune response.
Line 35 The LecRLKs are believed to be involved in plant immune receptors. It is not clear what does it means
Please delete from line 40 to 43 and try to provide more general information since potato is cultivated wordwide
Delete from line 46-50
From line 51 to line 56 The goals of work shoud be better delineated
Modifiy Methods in Material and methods
From line 59 to line 66 Provide clear information about data downloaded, proteome versions used and matching repository
67 change were obtained as method in following the method
79 Please better define what you consider a cluster since this is a very tricky point
You can use a window approach or a gap approach (Richly et al. 2002; Luo et al 2012; Di Donato et al 2017)
92 transformed may be is transferred
Results
116 Rise may be is Rice
Please comment table 1 or delete it
118 Proteome were downloaded from different repositories and we have not information on version used as well as sequencing and assembling methods. So you should be wise in presenting data. Moreover, you cannot compare data form literature obtained with different procedures. Please modify description according to this point
From line 135 to 138 Too many details
From 142 to 143 I’ don’t think that data support such finding
Line 174 It is not clear
Line 176 Like above three proteins located in the same position of potato genome and only one of these was visualized on chromosome. Proteins on chromosomes ????
Figure Legends should be better detailed

Discussion should be carefully revised.
For instance, you cannot write at line 326 the proteins could be used for further resistant breeding.
You just demonstrated that is active during plant-pathogen interaction

Reviewer 2 ·

Basic reporting

The work of Zhang and Qin presents an in silico genome-wide analysis of the lectin receptor-like kinases in the crop plant Solanum tuberosum. Authors follow up by looking at the expression of selected members of the family in plants challenged with bacterial and fungal infection to identify candidates for further studies. I find the work both interesting and relevant. The aims are clearly explained. As stated above experimental strategy is well described and follows an established route.

Nevertheless I find that the manuscript could be further improved.

Major revisions

1. The introduction would benefit from referring to the closely related and published work of Nazarian-Firouzabadi F et al., 2019 (doi: 10.1007/s11033-019-04951-z); Li et al., 2018 (doi: 10.3390/cells7090120); Dezhsetan S. 2018 (10.1007/s12298-017-0471-6). Along similar lines I suggest that authors discuss how their work compares to the published studies (see above) dealing with in silico identification of receptor-like kinases in Solanum tuberosum.

2. The English language should be improved. However, the text is understandable; it suffers from grammatical and stylistic errors. For instance, and starting with the first sentence of the abstract “commonly discovered” would be better phrased as “evolutionary conserved in the plant lineage,” the verb is missing before “involved in.” There are multiple examples throughout the text, including statements like the one in line 119 /120. Moss is also a plant; although non-vascular.

3. I thank the authors for providing the raw data. However, they would greatly benefit from a better description. As it stands now, I found it challenging to extract information. For instance, both the “q-RT-PCR terminal data” and “LEC-Zhang” files lack column headers.

Minor revisions

Figures 5 and 6: Could the authors refer to publication and/or depository from where they extracted the data?
Lines 195 and 201. It would be great if the authors could specify how they define differential expression.
Line 226. Could the authors justify selection of these particular 14 genes?

Experimental design

As stated above experimental strategy is well described and follows an established route. However I find an issue with an experimental design.

Major point

4. Infection experiments (Figure 7 and Figure 8) were performed three times; this is how I understand the meaning of three bio-reps (lines 236 and 246). Could the authors provide more information? Do they refer to the three completely independent experiments or for example, separate leaves/tubers in one experiment? It is not clear to me why having three independent biological experiments; the authors decided to merge the material and rely on three technical measurements of the same cDNA sample (Lines 108 and 109). I gather that the bar plots on figures 7 and 8 represent mean ± SD. Am I correct? This information should be in the legend. Performing statistical analysis and discussing significance, having exclusively technical replication is highly problematic. I suggest that the authors repeat the expression analysis and provide data for the three biological replications.

Validity of the findings

Please see my comment to the experimental design.

---

## Round 0.2 · Major Revisions

The article requires additional revision in accordance with the recommendations of the reviewer.

Reviewer 2 ·

Basic reporting

See in the General comments for the author.

Experimental design

See in the General comments for the author.

Validity of the findings

See in the General comments for the author.

Additional comments

I find the manuscript of Zhang and Qin improved. Nevertheless, there are still points that need addressing.

Authors did not refer to the suggested literature, which is fine, if they explained why rather than writing “We have revised them in the manuscript” when they did not.

For the English editing, it could be done better as I can still find plenty of grammatical, stylistic, and punctuation errors. For instance, the second sentence in the abstract should be “largely” and not “large.”

Authors used “an absolute value of log2 ratio ≥ 1 as thresholds to judge the significance of differences in gene expression” (lines 88 and 89). Why use a fold change cut-off rather than a statistical test? Are reported changes truly significant?

I am grateful to the authors for explaining replication. An extra line (or two) in Materials in Methods explaining what constitutes a biological replica is necessarry (also for the readers to know).

I am grateful to the authors for providing headers in the Tables S1 to S4. Now I can understand the data, I have few questions to table S3. Why are the raw Ct values given for either mock or treatment, but never both? Why did the authors decide to use fold change rather than deltaCt values to calculate TTEST? I am a bit surprised that actin Ct values are always so close between three replicas of the same time point/treatment. For the measurements done on independent cDNA preparation, I would expect random distribution of the actin values, rather than rep 1 to 3 clustering together. Could the authors think of the undelaying reason? Small comment, there are spelling errors in the headers, please correct. For instance, in Table S3 should be “with” rather than “ith”.

---

## Round 0.3 · Major Revisions

The authors have provided an improved manuscript and a detailed explanation of all changes done.

Gerard Lazo, the Section Editor, has commented and said:
>
> "I applaud the authors on what they characterize as a deep analysis of the lectin RLK family and successfully analyze and attempt classification based on several parameters; yet the information does not present it in a fashion that can be readily recognized and moved forward by the readership. I would expect some ontology classifications based on the analysis of 113 member for the species studies which was broken down into 7 clade groups; and on top of that were pointers to specific expression against stress challenge. There are many factorial analyses which can be added. Considering that this was open data and that many common pipelines of analysis were used, one more would not hurt and would add utility. As the source data was readily downloaded and analyzed I would expect the authors to provide connections to the sequence data as it is not readily retrieved from the information provided; it appears only based on name only without coordinates from the source material, or even default FASTA sequences of the candidates analyzed.
>
> Journal manuscripts are often scanned by text-mining software that locates and extracts core data elements, like gene function. Adding standard ontology terms, such as the Gene Ontology (GO, geneontology.org) or others from the OBO foundry (obofoundry.org) can enhance the recognition of your contribution and description. This will also make human curation of literature easier and more accurate. None of this was visible. Such features might be added to one of the tables, especilly for the Solanum genomes, as other species were also included, but not the direct subject matter of the manuscript.
>
> Once this request is provided I will gladly approve this being moved forward, I would place this as a major modification needed."

---

## Round 0.4 · accepted · Accept

The authors correctly addressed the reviewer comments and suggestions and detailed these changes in the manuscript.